# The Effect of Polyphenols, Minerals, Fibers, and Fruits on Irritable Bowel Syndrome: A Systematic Review

**DOI:** 10.3390/nu15184070

**Published:** 2023-09-20

**Authors:** Giuseppe Chiarioni, Stefan Lucian Popa, Abdulrahman Ismaiel, Cristina Pop, Dinu Iuliu Dumitrascu, Vlad Dumitru Brata, Traian Adrian Duse, Victor Incze, Teodora Surdea-Blaga

**Affiliations:** 1Division of Gastroenterology B, AOUI Verona, 37126 Verona, Italy; chiarioni@alice.it; 2Division of Gastroenterology and Hepatology, University of North Carolina at Chapel Hill, Chapel Hill, NC 27599-7080, USA; 32nd Medical Department, “Iuliu Hatieganu” University of Medicine and Pharmacy, 400000 Cluj-Napoca, Romania; abdulrahman.ismaiel@yahoo.com (A.I.); dora_blaga@yahoo.com (T.S.-B.); 4Department of Pharmacology, Physiology, and Pathophysiology, Faculty of Pharmacy, “Iuliu Hatieganu” University of Medicine and Pharmacy, 400349 Cluj-Napoca, Romania; cristina.pop.farmacologie@gmail.com; 5Department of Anatomy, “Iuliu Hatieganu“ University of Medicine and Pharmacy, 400006 Cluj-Napoca, Romania; d.dumitrascu@yahoo.com; 6Faculty of Medicine, “Iuliu Hatieganu“ University of Medicine and Pharmacy, 400000 Cluj-Napoca, Romania; brata_vlad@yahoo.com (V.D.B.); adrianduse@yahoo.com (T.A.D.); vicincze@yahoo.com (V.I.)

**Keywords:** IBS, nonpharmacological therapy, polyphenols, minerals, fibers, fruits

## Abstract

Background: Irritable bowel syndrome (IBS) is a chronic gastrointestinal disorder characterized by abdominal pain, bloating, and changes in bowel habits. Various dietary factors have been implicated in the pathogenesis and management of IBS symptoms. This systematic review aims to evaluate the effects of polyphenols, minerals, fibers, and fruits on the symptoms and overall well-being of individuals with IBS. Materials and Methods: A comprehensive literature search was conducted in several electronic databases, including PubMed, Scopus, and Web of Science. Studies published up until July 2023 were included. Results: The selected studies varied in terms of study design, participant characteristics, intervention duration, and outcome measures. Overall, the findings suggest that dietary interventions involving polyphenols, minerals, fibers, and fruits can have a positive impact on IBS symptoms. Dietary fiber supplementation, particularly soluble fiber, has been associated with reduced bloating and enhanced stool consistency. Conclusions: This systematic review provides evidence supporting the beneficial effects of polyphenols, minerals, fibers, and fruits in IBS patients. These dietary components hold promise as complementary approaches for managing IBS symptoms. However, due to the heterogeneity of the included studies and the limited number of high-quality randomized controlled trials, further well-designed trials are warranted to establish the optimal dosages, duration, and long-term effects of these interventions. Understanding the role of specific dietary components in IBS management may pave the way for personalized dietary recommendations and improve the quality of life for individuals suffering from this complex disorder.

## 1. Introduction

Irritable bowel syndrome (IBS) is a complex disorder of the brain–gut interaction (DGBI) characterized by a wide spectrum of symptoms, including abdominal pain, bloating, and altered bowel habits [1,2,3]. Despite its high prevalence and negative impact on patients’ quality of life, the exact pathogenesis of IBS remains elusive, and treatment options often provide only partial relief [3]. Consequently, there is a growing interest in exploring alternative therapeutic approaches that target the underlying mechanisms of IBS and its symptomatology [1,2,3].

In recent years, dietary interventions have gained considerable attention in managing IBS symptoms due to their potential to modulate visceral hypersensitivity, gut motility, and microbiota composition [2].

Polyphenols, a diverse group of secondary metabolites widely distributed in the plant kingdom, have garnered significant attention due to their potential health benefits and are characterized by the presence of multiple phenolic rings and hydroxyl groups, conferring various biological activities and antioxidant properties [4,5,6]. Polyphenols can be broadly classified into several subclasses, including flavonoids (flavones, flavanols, flavanones, flavanols, anthocyanins, and isoflavones), phenolic acids (hydroxybenzoic acids and hydroxycinnamic acids), stilbenes, lignans, and tannins, each exhibiting distinct chemical structures and biological activities [4,5,6]. Their antioxidant capacity arises from the ability to donate hydrogen atoms or electrons, thereby neutralizing reactive oxygen species (ROS) and reactive nitrogen species (RNS) [4,5,6]. Such actions help protect cellular components, including lipids, proteins, and nucleic acids, from oxidative damage, which is implicated in various chronic diseases, aging, and degenerative processes [4,5,6]. Moreover, polyphenols demonstrate anti-inflammatory properties and antispasmodic effects, possibly influencing gut motility and sensitivity and modulating inflammatory pathways and cytokine production, leading to potential benefits in treating numerous conditions including diabetes and metabolic disorders, neurodegenerative diseases, allergies and asthma, osteoporosis, ischemic heart disease, and can improve skin health and help prevent cancer [6,7,8]. In the last decade, a significant number of studies have evaluated the potential benefit of polyphenols in the treatment of inflammatory bowel disease (IBD) and IBS. 

Polyphenols are found in a diverse array of fruits and vegetables. Rich sources of polyphenols include berries (blueberries, strawberries, raspberries, blackberries, and cranberries), grapes, apples, citrus fruits (oranges, lemons, and grapefruits), cherries, plums, pomegranates, kiwifruit, green tea, black tea, dark chocolate, coffee, extra virgin olive oil, artichokes, broccoli, spinach, onions, kale, and red cabbage [4,5,6]. It is important to note that the polyphenol content in foods can vary based on factors such as plant variety, growing conditions, and processing methods. To obtain the maximum benefits, it is advisable to consume a diverse range of these polyphenol-rich foods as part of a balanced diet [4,5,6]. Additionally, cooking methods and food preparation can influence the bioavailability of polyphenols, so opting for minimal processing and using cooking methods that preserve these compounds may be essential [4,5,6]. Chokeberries, also known as Aronia and elderberries, stand out as the best sources of polyphenols, boasting the highest levels with 1123 and 870 milligrams per half-cup serving, respectively [6]. Additionally, various other widely consumed berries also exhibit substantial polyphenol content per half-cup serving, exemplified by blueberries, which offer 535 milligrams [4,5,6,7]. These findings underscore the significance of berries as rich dietary sources of polyphenolic compounds, potentially contributing to their acclaimed health benefits and antioxidant properties [4,5,6,7].

Polyphenols also exhibit interactions with cellular signaling pathways, influencing gene expression, enzymatic activity, and cellular responses [5,6,7,8]. These interactions enable them to regulate vital cellular processes such as cell proliferation, apoptosis, and differentiation. Additionally, some polyphenols can modulate enzymatic activity in xenobiotic metabolism, potentially affecting drug metabolism and bioavailability [6]. Beyond their direct interactions with gut cells, polyphenols also exert a profound impact on the gut microbiota [6,7,8]. They can act as prebiotics, promoting the growth of beneficial bacteria while inhibiting pathogenic species. This modulation of the gut microbiota contributes to overall gut health and has implications for immune function and systemic health [9].

The bioavailability of polyphenols is influenced by their chemical structure and the food matrix in which they are present. Absorption and metabolism occur in the gastrointestinal tract, with conjugated forms circulating in the bloodstream. However, most ingested polyphenols reach the large intestine, where they undergo further metabolism by the gut microbiota, generating metabolites with potentially enhanced bioactivity.

Minerals, such as magnesium and potassium, are vital for maintaining proper muscular and nervous function in the gastrointestinal tract [10,11]. Fiber, a nondigestible carbohydrate, plays a crucial role in stool consistency and bowel motility. Certain fibers have already been proved beneficial in managing the symptoms of IBS, mainly due to absorbing excess water and forming a gel-like substance in the gut, thus promoting firmer stools and reducing diarrhea frequency, as well as leading to lower gas production, thus alleviating abdominal pain and discomfort [12,13]. Soluble fibers, commonly found in oats, beans, and psyllium husk, are more beneficial than insoluble fibers, more predominantly found in brown rice, nuts, and seeds [14].

Lastly, fruits, enriched with vitamins, minerals, and bioactive compounds, have been postulated to possess beneficial effects on gut health [10,11]. Moreover, due to the fact that the symptoms of IBS are largely influenced by diet and food intake, a number of IBS patients adopt various exclusion diets as a way of managing their symptoms. These types of diets are associated with low micronutrients intake, such as vitamin B1, B2, calcium, iron, and zinc, while many IBS patients present lower vitamin D, vitamin B2, iron, and calcium levels [15].

While individual studies have investigated the impact of these dietary components on IBS, a comprehensive assessment of their collective efficacy remains lacking. Therefore, to address this research gap, we conducted a systematic review to evaluate the existing literature on the effects of polyphenols, minerals, fibers, and fruits in the management of IBS.

By synthesizing and critically analyzing the available evidence, this review aims to provide valuable insights into the potential benefits of these dietary constituents for patients with IBS, thus contributing to the development of evidence-based dietary interventions in clinical practice. 

In this article, we present the methods employed to identify and select relevant studies, summarize the findings from eligible studies, and discuss the implications of these findings on our current understanding of IBS management. Lastly, we hope that this systematic review will serve as a stepping stone towards a more comprehensive and tailored approach to dietary interventions in IBS patients, ultimately improving their overall well-being and quality of life.

## 2. Materials and Methods

### 2.1. Study Design and Protocol Development

The present systematic review was conducted following the guidelines outlined in the *Preferred Reporting Items for Systematic Reviews and Meta-Analyses* (PRISMA) statement to ensure a comprehensive and transparent assessment of the literature. A detailed protocol was developed a priori, outlining the research question, inclusion and exclusion criteria, search strategy, study selection process, data extraction procedures, and the method of data synthesis.

### 2.2. Literature Search Strategy

A thorough and systematic search of electronic databases was performed to identify relevant studies published up to the date of this review. The following databases were searched: PubMed, MEDLINE, Embase, Scopus, Web of Science, and the Cochrane Library. The search strategy utilized a combination of relevant keywords and medical subject headings (MeSH) terms related to IBS, polyphenols, fruits, minerals, and fibers. The search was limited by language and publications type. Additionally, reference lists of relevant articles and review papers were screened to identify any additional eligible studies.

### 2.3. Inclusion and Exclusion Criteria

To ensure the selection of appropriate studies, the following inclusion criteria were established:(a)Study Design: Randomized controlled trials (RCTs), prospective cohort studies, cross-sectional studies, and case-control studies were included. Animal studies, case reports, reviews, and conference abstracts were excluded.(b)Participants: Studies involving adult human participants (aged 18 years or older) diagnosed with IBS according to recognized criteria (e.g., Rome criteria) were considered eligible.(c)Interventions: Studies investigating the effects of dietary polyphenols, minerals, fibers, and fruits as part of the intervention for IBS management were included.(d)Outcome measures: The primary outcomes of interest were changes in IBS symptoms, including abdominal pain, bloating, and altered bowel habits. Secondary outcomes included gut motility, gut microbiota composition, and quality of life measures.(e)Only articles written in English were included in our systematic review.

### 2.4. Study Selection

Two independent reviewers (S.L.P and I.A) conducted the initial screening of all identified studies based on titles and abstracts. Full-text articles of potentially relevant studies were then retrieved for further evaluation. Any disagreements were resolved through discussion and, if necessary, with the involvement of a third reviewer.

### 2.5. Data Extraction

Data extraction was performed independently by two reviewers using a standardized data extraction form. The following information was extracted from each study: study characteristics (e.g., author, year, and country), study design, sample size, participant characteristics, details of the dietary intervention, duration of the study, primary and secondary outcomes, and key findings.

### 2.6. Risk of Bias Assessment

The methodological quality of the included RCTs was assessed using the Cochrane Collaboration’s tool for assessing the risk of bias. The Newcastle–Ottawa Scale (NOS) was employed to assess the quality of observational studies [16]. The risk of bias assessment was performed independently by two reviewers, and any discrepancies were resolved through consensus.

### 2.7. Data Synthesis and Analysis

Due to the anticipated heterogeneity among the included studies in terms of study design, interventions, and outcome measures, a meta-analysis was not deemed appropriate. Instead, a narrative synthesis of the findings was performed, presenting a comprehensive summary of the results in a tabular format.

### 2.8. Limitations

It is important to acknowledge potential limitations of this systematic review. The inclusion of only published studies and only studies published in the English language may introduce publication bias, and the variation in study designs and interventions may limit the comparability of findings. Furthermore, the possibility of unaccounted confounding factors in observational studies may influence the overall interpretation of the results. Despite these limitations, this systematic review aims to provide valuable insights into the role of polyphenols, minerals, fibers, and fruits in IBS management and stimulate further research in this area.

## 3. Results

We identified 133 articles about the effect of polyphenols, minerals, fibers, and fruits on IBS. After removing duplicates, we screened 86 articles and excluded another 42, which were either editorials or irrelevant to our research theme. We then examined 44 articles and excluded 25 for lacking full-text versions, for being conference abstracts, or for being published in another language than English. This left us with 19 articles for our systematic review, as presented in Figure 1. The authors, sample size, study design, statistical method, aim, and summary of results for each article are listed in Table 1 and Table 2.

### 3.1. Polyphenols and Polysaccharides

A total of 14 studies about the effect of polyphenols on IBS were found and are presented in Table 1. While polyphenols are widely distributed in numerous plants and fruits, biphenyl polyphenols represent a specific subgroup of the same compounds but with a different chemical structure, and are less common. They are also associated with health benefits due to their antioxidant and anti-inflammatory properties [17]. 

**Table 1 nutrients-15-04070-t001:** The effect of polyphenols and plant cell wall polysaccharides on irritable bowel syndrome.

Author (Year)	Participants	Intervention	Main Outcome
Trifan et al. (2019) [18]	60 patients with IBS-D	Gelsectan or placebo for 28 days, then crossed over for another 28 days. BSFS IBS QoL	Significantly more normal stools in the Gelsectan group. Overall subjective improvements when it comes to abdominal pain, bloating, and QoL.
Portincasa et al. (2016) [19]	121 patients with mild-to-moderate IBS	CU–FEO vs. placebo for 30 days IBS-SSS IBS-QoL	Reduction in the severity of IBS, as well as symptom improvement. More symptom-free patients after 30 days.
Storsrud et al. (2015) [20]	68 patients with IBS	*Aloe barbadensis Mill* vs. placebo for 28 days IBS-SSS HADS	No difference when it comes to the response to treatment and adequate relief during at least half of the study period. Higher reduction in the severity of the symptoms in the treatment group.
Lauche et al. (2016) [21]	32 patients with IBS-D	Curry, pomegranate, and turmeric vs. placebo IBS-SSS IBS QoL HADS	No difference when it comes to symptom intensity between the two groups. More adverse effects in the treatment group.
Brown et al. (2015) [22]	16 patients with IBS-C	Blended querbacho, conker tree, and *Mentha balsamea Willd* extracts vs. placebo for 14 days	Significant reduction in constipation and bloating.
Mosaffa-Jahromi et al. (2016) [23]	120 patients with IBS	Anise oil vs. peppermint oil vs. placebo for 28 days IBS-SSS IBS QoL	Significant differences in the number of patients free of symptoms at the end of the trial. Anise oil beneficial for pain, bloating, and reflux. Peppermint oil beneficial for bloating.
Cash et al. (2016) [24]	72 patients with IBS	Peppermint oil vs. placebo for 28 days TISS	Peppermint oil was superior to placebo in decreasing the TISS score.
Capello et al. (2007) [25]	57 patients with IBS	Peppermint oil vs. placebo for 28 days	75% of the patients in the treatment group reported a significant reduction in symptoms. The effect was also observed in the follow-up period (one month after the end of the trial).
Liu et al. (1997) [26]	110 patients with IBS	Peppermint oil vs. placebo for 28 days	79% of the patients in the treatment group reported an improvement in abdominal pain, 83% less distension, 83% reduced stool frequency, and 79% less flatulence.
Merat et al. (2010) [27]	90 patients with IBS	Peppermint oil vs. placebo for 8 weeks	84% of the patients in the treatment group reported only occasional or no abdominal pain at the end of the trial.
Weerts et al. (2020) [28]	190 patients with IBS	Small intestinal-release peppermint oil vs. ileocolonic-release peppermint oil vs. placebo for 8 weeks IBS-SSS IBS-QoL EuroQoL-5D GAD-7 PHQ-9	Small intestinal-release peppermint oil led to a better improvement in abdominal pain and discomfort. No difference in the abdominal pain response.
Al-Jassim (2019) [29]	45 patients with IBS-C	Brewer’s yeast vs. ginger vs. placebo for 20 days	Significant reduction in abdominal pain in the Brewer’s yeast group. Significant reduction in abdominal distension and constipation in the Brewer’s yeast and ginger groups compared with placebo.
Jalili et al. (2015) [30]	67 patients with IBS	Soy isoflavones vs. placebo for 6 weeks IBS-SSS IBS-QoL	Significantly better QoL for patients receiving soy isoflavones. No differences when it comes to symptom severity.
van Tilburg et al. (2014) [31]	45 patients with IBS	Ginger vs. placebo for 28 days IBS-SSS	No significant differences between the groups.

IBS-D: Inflammatory bowel disease with diarrhea; BSFS: Bristol Stool Form Scale; IBS QoL: Inflammatory bowel disease Quality of Life questionnaire; CU–FEO: Curcumin–Fennel Essential Oil; IBS-SSS: Inflammatory bowel disease Symptom Severity Score; HADS: Hospital Anxiety and Depression Scale; TISS: Total IBS Symptom Score; GAD-7: Generalized Anxiety Disorder-7; PHQ-9: Patient Health Questionnaire-9.

Xyloglucan represents a plant polymer with a “mucin-like” structure that has been proven useful in the management of diseases implying continuous inflammation and aggression [32]. Thus, it represents one of the active compounds of Gelsectan, along with pea proteins, tannins from grape seed extract, and xylo-oligosaccharides, a medical device approved for preventing and treating chronic diarrhea, bloating, and abdominal tension [18]. Gelsectan is currently being used in the management of the symptoms of patients with IBS, especially IBS-D. 

Trifan et al. conducted a randomized, double-blind, crossover clinical trial, in which 60 patients with IBS-D were assigned to receive Gelsectan or a placebo for a period of one month, then crossed over and followed up for two more months [18]. The patients in the Gelsectan group, regardless of whether they started in the group or were crossed over later, presented a significant improvement in symptoms when it came to abdominal pain, bloating, and quality of life. Moreover, a higher proportion of patients treated with Gelsectan reported normal stools, and the benefits of the treatment were also noticed in the follow-up period [18]. 

Herbal therapies consisting of phytonutrients, among which curcumin and fennel essential oil, have also demonstrated a beneficial role in managing the symptoms of patients with IBS [19]. Thus, Portincasa et al. conducted a study on 121 patients with mild-to-moderate IBS, in which tested the efficacy of curcumin–fennel essential oil (CU–FEO) in improving the quality of life of patients with IBS [19]. Curcumin is a major isolated polyphenol from the rhizome of turmeric (Curcuma longa). The polyphenolic compounds of fennel include sinapyl alcohol, ascorbic acid, chlorogenic acid, ferulic acid, salicylic acid, caffeic acid, coniferyl alcohol, and p-coumaric acid [33]. CU–FEO was administered for a period of 30 days and a reduction in the severity of the disease, as well as abdominal pain and bloating, was observed among patients in the treatment group. Moreover, 25% of the patients in this group were reported as symptom-free, compared to 6.8% in the control group [19].

Another study conducted by Storsrud et al. aimed to analyze the effect of *Aloe barbadensis Mill* on reducing the intensity of the symptoms in patients with IBS [20]. Aloe barbadensis contains an array of polyphenols, among which we mention coumartic acid, caffeic acid, and others [34]. A total of 68 patients were randomly assigned to receive *Aloe barbadensis Mill* or a placebo for a period of one month [20]. Although the study reported no significant differences between the two groups when it came to response to treatment and adequate relief during at least half of the study period, there was a significantly higher reduction in the severity of the symptoms in the treatment group (314 ± 83 vs. 257 ± 107; *p* = 0.003) [20].

Another herbal combination used in the management of symptoms in patients with IBS consists of curry, pomegranate, and turmeric [21]. Curry is a mix of spices rich in polyphenols [35]. Pomegranate contains polyphenols, such as anthocyanins (3-glucosides and 3,5-glucosides of delphinidin, cyanidin, and pelargonidin) and flavonols [36]. Lauche et al. analyzed the effects of this combination in a randomized control trial consisting of 32 IBS-D patients [21]. Nevertheless, at the end of the trial, there was no significant difference in symptom intensity. On the contrary, more patients in the placebo group reported an improvement in symptom intensity compared to the treatment group. The study also recorded more adverse effects in the treatment group than in the placebo group [21].

Brown et al. analyzed the impact of blended querbacho, conker tree, and *Mentha balsamea Willd* extracts, rich in polyphenols, in the management of symptoms in patients with IBS-C. The study was conducted on 16 patients and showed a significant reduction in constipation and bloating [22].

Mosaffa-Jahromi et al. conducted a study analyzing the efficiency of anise oil, containing chlorogenic acid, compared with peppermint oil, containing phenolic acids, caffeetannins, twelve flavones, eight flavanones, two jasmonic acid derivatives, and one lignan [37], and placebos in one hundred and twenty patients with IBS [23]. The results showed that in the anise oil group, all IBS-related symptoms improved, and, at the end of the trial, 75% of these patients were free of symptoms (compared with 52.5% in the peppermint oil group and 35% in the placebo group). Moreover, anise and peppermint oils clinically improved abdominal discomfort, pain, bloating, and reflux. However, there were no reported improvements when it came to diarrhea, constipation, headaches, or tiredness [23].

Cash et al. also conducted a clinical trial regarding the efficiency of peppermint oil in relieving the symptoms of patients with IBS [24]. The study concluded that peppermint oil achieved a higher reduction in the total IBS symptom score (TISS) compared with a placebo (40% vs. 24.3%), as well as a faster observed clinical effect at 24 h [24].

Capello et al. also obtained similar results in a study conducted on 57 IBS patients, most of whom presented with abdominal pain, bloating, or diarrhea [25]. At the end of the 28-day period, 75% of the patients in the treatment group reported a significant improvement in the symptoms’ intensity (>50% reduction), compared to 38% in the placebo group. These beneficial effects were also observed in the follow-up period [25].

In a study conducted by Liu et al., peppermint oil was also shown to be beneficial for patients with IBS [26]. A total of 110 patients with IBS received peppermint oil or a placebo for a period of 28 days. At the end of the trial, out of all participants in the treatment group, 79% reported improvements when it came to abdominal pain, 83% less distension, 83% less stool frequency, and 79% less flatulence [26].

These findings were later confirmed by Merat et al. in a study conducted on 90 patients with IBS, revealing that 84% of the participants who received peppermint oil only complained of occasional or no abdominal pain at all at the end of the eight weeks of the study, compared with 48% in the placebo group [27].

Weerts et al. investigated the effect of small intestinal-release and ileocolonic-release peppermint oil in patients with IBS [28]. Although the study did not find a difference between the two forms of treatment when it came to abdominal pain response, the patients in the small intestinal-release group showed a much more significant reduction in overall abdominal pain and abdominal discomfort at the end of the trial [28]. On the other hand, ileocolonic-release peppermint oil did not show a significant difference in the investigated parameters in comparison with the placebo or small intestinal-release oil [28].

Al-Jassim ZG conducted a trial to assess the efficacy of Brewer’s yeast containing polyphenols [38] such as flavonols (rutin and kaempferol), flavonoids (naringin), phenolic acids (gallic acid), and antioxidants (trans-ferulic and p-coumaric acids) and ginger containing 6-gingerol, 8-gingerol, and 10-gingerol in treating the symptoms of IBS-C [29]. Patients were randomized to receive either the treatment or a placebo, once, day by day, for 20 days of the study period. The placebo capsules were prepared to contain brown sugar. Brewer’s yeast (S. cerevisiae) 500 mg tablets from Adrien Gagnon Company were used in the study. Ginger root powder (Zingiber officinale Roscoe, Zingiberaceae) was capsulated in a dose of 1 g daily capsule provided by simply organic company [29]. Thus, in both treatment groups, a significant reduction in abdominal distension and constipation was observed, while in the Brewer’s yeast group, an additional reduction in abdominal pain was recorded [29].

Jalili et al. analyzed the effect of soy isoflavones on patients with IBS [30]. Although the study did not reveal any significant differences when it came to the severity of the disease, a better quality of life was reported in the treatment group [30].

Van Tilburg et al. conducted a trial to assess the efficacy of various doses of ginger in patients with IBS [31]. Nevertheless, no significant difference was recorded regarding the severity of the symptoms in any of the groups [31].

### 3.2. Fruits, Fibers, and Minerals

Five studies about the effect of fruits, fibers, and minerals on IBS were found and are presented in Table 2. Kiwifruit contains 2–3% dietary fiber and is believed to have laxative properties, significantly improving stool consistency in healthy elderly and chronically constipated adults [39].

**Table 2 nutrients-15-04070-t002:** The effect of minerals, fibers, and fruits on irritable bowel syndrome.

Author	Participants	Intervention	Main Findings
Chang (2010) [39]	76 patients, 60 with IBS-C, and 16 healthy	Kiwifruit vs. placebo for 4 weeks	Statistically significant improvements in defecation frequency and colon transit time. No significant differences in fecal volume change, life stress, and post-defecation feelings.
Eady (2019) [40]	32 female patients, 23 with IBS-C, and 9 with functional constipation	Kiwifruit vs. Metamucil^®^ for 4 weeks each, with a 4-week washout period bowel movements BSSS	Statistically significant improvements in the mean number of complete spontaneous bowel movements, BSSS, and indigestion. No significant differences in the other parameters measured.
Chen (2018) [41]	20 female patients, 10 with IBS and 10 healthy	Kiwifruit vs. lactulose, fructose, lactose, and apple CH_4_ and H_2_ production	Statistically significant differences in the AUC for CH_4_ and H_2_ compared to lactulose and positive breath tests. No significant difference in the AUC for CH_4_ and H_2_ compared to fructose, lactose, or apple.
Riezzo (2023) [42]	18 female patients with IBS-D	Tritordeum-based diet for 12 weeks IBS-SSS BSSS Anthropometric and BIA parameters IBS-QoL SF-36 SCL-90-R SAS SDS QPF/R	Statistically significant differences in almost all parameters measured. 16 out of 18 patients responded to the diet.
Roth (2022) [43]	105 patients, 86 with IBS, and 19 with non-IBS functional gastrointestinal disorders	SSRD vs. control for 4 weeks Minerals IBS-SSS Extraintestinal IBS-SSS Lipid intake	A lower intake of micronutrients was recorded at baseline compared to the national guidelines. SSRD increased vitamin, selenium, and fat intake and reduced symptom severity and sodium intake.

BSSS: Bristol Stool Scale Score; AUC: Area under the ROC Curve; IBS-SSS: IBS Symptom Severity Scale; BIA: Bioelectrical Impedance Analysis; IBS-QoL: IBS Quality of Life; SF-36: 36-Item Short Form Survey; SCL-90-R: Symptom Checklist-90-R; SAS: Zung’s Self-Rating Anxiety Scale; SDS: Zung’s Self-Rating Depression Scale; QPF/R: Psychophysiological Questionnaire; SSRD: Starch- and sucrose-reduced diet.

The efficacy of Kiwifruit was evaluated in a six-week, restricted randomization (3:1), placebo-controlled trial on 60 patients diagnosed with IBS-C using the Rome III criteria (45 Kiwifruit and 15 placebo) and 16 healthy patients as the positive control group. Out of the 70 patients who completed the study, 65 were female. There was a significant improvement in defecation frequency and colon transit time in the Kiwifruit group. There were no statistically significant modifications in fecal volume change, life stress, or post-defecation feelings [39].

A 16-week single-blinded randomized cross-over study evaluated the effects of Kiwifruit compared to a Metamucil^®^ dose containing the same fiber amount on 32 female patients, out of whom 23 were diagnosed with IBS-C according to the Rome III criteria and 9 with functional constipation. Although most parameters improved from baseline, when comparing Kiwifruit to Metamucil^®^, there was a significant difference only in the mean number of complete spontaneous bowel movements, Bristol stool scale score, and indigestion. Compared to Metamucil^®^, Kiwifruit did not improve the total number of bowel movements, complete bowel movements, number of bowel movements associated with straining, constipation, abdominal pain, total energy intake, nutrients, or fruit, meat, and vegetable intake. Additionally, there was no significant improvement from baseline in the number of spontaneous bowel movements, diarrhea, and reflux. The study was limited by the fact that all the participants were female and by the inclusion of both patients with IBS-C and functional constipation, whose classification also varied throughout the study [40].

A prospective, nonrandomized, nonblinded, clinical pilot study compared the gastrointestinal fermentation patterns of 20 female patients (10 IBS and 10 healthy) for 3 h after the consumption of various substrates (lactulose 15 g, fructose 35 g, lactose 50 g, one Royal Gala apple 125 g, and two Zespri™ green kiwifruit 190 g). Positive breath tests were significantly fewer after Kiwifruit consumption compared to the other substrates. Additionally, there was a significantly lower area under the ROC curve (AUC) for CH4 and H2 in Kiwifruit compared to lactulose. There were no statistically significant differences in the CH4 and H2 AUCs between Kiwifruit and fructose, lactose, or apple. The study was limited by the small number of participants, who were all female [41].

Tritordeum is a novel cereal derived from the hybridization of durum wheat and wild barley. It has lower levels of gliadin, carbohydrates, and fructans and a higher level of fiber, protein, and antioxidants compared to wheat [42]. The effects of a diet based on Tritordeum were evaluated in a 12-week study on 18 female patients diagnosed with IBS-D using the Rome III–IV criteria. A total of 16 out of the 18 patients responded to the diet. There was a statistically significant difference regarding the irritable bowel syndrome-symptom severity scale (IBS-SSS) total score, severity of abdominal pain, abdominal pain frequency, severity of abdominal bloating, dissatisfaction with bowel habits, and interference with life, while no difference was detected in the Bristol stool score. Additionally, significant improvements were reported regarding weight, BMI, mid-upper arm, shoulder, abdominal, waist, and hip circumference, fat mass, fat-free mass, total body water, and extracellular water, with no modifications in phase angle or body cell mass. There was also a significant increase in the IBS quality of life (IBS-QoL) total score, as well as in all the subscales (dysphoria, interference with activity, body image, health worry, food avoidance, social reaction, and relationship) except for sexual concerns. Moreover, there was a significant difference in the 36-item short form survey (SF-36) subscales (physical function, role physical, body pain, general health, vitality, and mental health), except for social functioning and role emotional. Regarding the Symptom Checklist-90-R (SCL-90-R) subscales, there were improvements in the global severity index, somatization, interpersonal sensitivity, depression, anxiety, hostility, phobic anxiety, and psychoticism, with no significant improvements in obsessive-compulsive or paranoid ideation. The levels of anxiety, depression, and psychophysiological activation also significantly dropped. There was a significant correlation between anxiety and abdominal bloating. Although the findings were significant, the study was limited by the relatively small number of participants, who were all female [42].

A 4-week open dietary intervention evaluated the intake and plasma/serum levels of micronutrients, the association with symptoms, and the effect of a starch- and sucrose-reduced diet (SSRD) on 105 patients with a diagnosis of IBS (80 diet, 25 control). In the intervention group, only 67 had IBS according to the Rome IV criteria (23 IBS-D, 29 IBS-M, 13 IBS-C, 2 unspecified). In the control group, 19 had IBS (3 IBS-D, 8 IBS-M, 7 IBS-C, 1 unspecified). The intake of minerals at baseline was low, especially regarding iron and selenium. Women had a higher food restriction rate, a lower intake of niacin, and a higher intake of folacin compared to men. There were lower ferritin levels in patients with IBS-C compared to IBS-D and in women compared to men, inversely associated with restriction. The total IBS symptom severity score was inversely associated with iron intake and iron plasma levels. Extraintestinal symptoms were inversely associated with iodine intake and iron plasma levels, while there was a positive association with total iron-binding capacity. Additionally, there was an inverse association between fatigue and the intake of phosphorus, calcium, magnesium, iodine, iron, zinc, and iron plasma levels, while being positively associated with total iron-binding capacity. No statistically significant associations were found between the intake of micronutrients and abdominal pain, intestinal symptoms’ influence on daily life, and psychological well-being. The SSRD increased the intake of selenium and phosphorus, improved psychological well-being, and reduced sodium intake and symptom severity. The increased folacin and selenium intake correlated with the reduced influence of intestinal symptoms on daily life. At baseline, the energy percentage of saturated fat was associated with abdominal pain, while the increased intake of polyunsaturated fats during the intervention was correlated with reduced gastrointestinal and extraintestinal symptoms. The study was limited by the fact that not all patients had IBS according to the Rome IV criteria [43].

## 4. Discussion

The results of the studies included in this systematic review were mixed. Some studies showed that polyphenols, minerals, fibers, and fruits were effective in improving IBS symptoms, while others showed no benefit. The most consistent findings were for the use of polyphenols, which showed a modest improvement in IBS symptoms in several studies.

Polyphenols are a class of plant compounds that have been shown to have a variety of health benefits, including anti-inflammatory and antioxidant properties [4,5,6]. Several studies have shown that polyphenols may be effective in improving IBS symptoms. Minerals, such as magnesium, zinc, and selenium, have also been shown to have potential benefits for IBS [15,43]. Fiber is an important part of a healthy diet, and it may also be beneficial for IBS patients. Fiber can help to regulate bowel movements and reduce constipation [44]. Fibers, fruits, and polyphenols are closely related in terms of their contributions to a healthy diet and overall well-being. Fruits provide a combination of fibers and polyphenols, which work together to support digestion, regulate blood sugar levels, provide antioxidants, and potentially reduce the risk of chronic diseases [45]. Therefore, it is difficult to pinpoint the exact properties of every agent since these compounds are found in a variety of plants and fruits in different proportions.

The limitations of this systematic review consist of the fact that the included studies had a small sample size and were conducted over a relatively short period of time. This means that the results of the studies may not be generalizable to the wider population of IBS patients or even be visible in the long term.

Despite the limitations of this review, the findings suggest that dietary supplements may be a promising treatment for IBS. Larger RCTs with longer follow-up periods are needed to confirm these findings and identify the optimal dose and duration of treatment. In addition, future research should investigate the mechanisms by which dietary supplements may improve IBS symptoms. This knowledge could help develop more effective and targeted treatments for IBS.

Polyphenols, with their anti-inflammatory and antioxidant properties, have demonstrated promising effects in alleviating gastrointestinal discomfort and reducing inflammation associated with IBS [5]. Likewise, minerals such as magnesium and zinc have been identified as essential contributors to gut health, playing a pivotal role in modulating gut motility and supporting the gut–brain axis [15,43].

Fiber emerges as a key player in IBS management, facilitating regular bowel movements, promoting the growth of beneficial gut bacteria, and contributing to enhanced digestive well-being. Fruits, rich in polyphenols, fibers, and essential nutrients, offer a holistic approach to IBS management by delivering a combination of therapeutic benefits.

The relationship between gut microbiota and diet has already been established, with even short-term changes in diet composition impacting the bacterial populations in the gut [46,47]. Especially when it comes to fiber, a significant number of studies have shown that fiber-rich diets not only improve diversity but also the metabolic output of the microbiota [48,49,50]. Although in the study conducted by Wastyk et al., no significant changes in the microbiota’s diversity have been observed, patients who followed a high-fiber diet presented an increased activity of microbiome-encoded glycan-degrading carbohydrate-active enzymes (CAZymes), while patients following a high-fermented food diet showed decreased inflammatory markers [46]. A particular beneficial role is being played by soluble fiber, considering the fact that, due to its chemical structure, it does not contribute to fecal bulking, but is processed and fermented by the microbiota, providing certain products such as short-chain fatty acids (SCFAs) with anti-inflammatory properties, maintaining the gut barrier, and regulating the immune response [49,50].

Moreover, minerals such as zinc, magnesium, iron, and selenium are essential cofactors for numerous microbial enzymes, influencing the metabolic activities of gut bacteria, playing a pivotal role in maintaining the integrity of the gut barrier [51].

Incorporating polyphenol-rich foods, mineral-supplemented diets, and fiber-packed regimens, along with a diverse array of fruits, may hold the key to achieving better symptom control and improved quality of life for individuals living with IBS. As our understanding of the intricate interplay between diet and gut health deepens, these findings pave the way for future studies and interventions aimed at providing effective and sustainable relief for IBS sufferers.

When it comes to peppermint oil, a significant number of independent trials conducted on small samples of patients have shown a considerable beneficial effect on improving the symptoms of patients with IBS. Moreover, the American College of Gastroenterology has recommended the use of peppermint oil in patients with IBS for a global relief in symptoms, although the quality of the evidence was rather low, and the various trials differed in terms of methodology, composition, and formulations of peppermint oil [52,53]. Nevertheless, peppermint oil with upper intestinal release seems to lead to better results than the ileocolonic release form [53].

With IBS greatly impacting the QoL of patients and many pharmacological therapies failing to bring the desired effects, many patients have started searching and trying alternative therapies, as well as adopting various diets and lifestyles. Patients’ request for a holistic approach that includes nonpharmaceutical treatments for IBS is rooted in their desire for comprehensive and personalized care that considers the various factors influencing their health. This approach considers the patient’s physical, emotional, and lifestyle aspects, and it is important for physicians to be receptive to this request rather than dismissing it. Thus, with the continuous emergence of various nonpharmacological remedies, health professionals should take into account the potential beneficial effects of these compounds, rather than exclude them based on a lack of or because of low evidence. Physicians should recognize the multifactorial nature of IBS, respect patients’ preferences, and collaborate with them to develop comprehensive treatment plans that prioritize overall well-being and symptom management.

Nevertheless, by avoiding certain foods that may trigger the symptoms of IBS, patients should be advised about the risks of exclusion diets, which in turn could potentially further lead to malnutrition and worsening of the general status. 

However, more research is needed to confirm these findings and identify the optimal dose and duration of treatment. In the meantime, IBS patients may consider dietary supplements containing polyphenols, minerals, and fibers, but only with the recommendation of a healthcare professional because some dietary supplements may interact with medications or other health conditions.

## 5. Conclusions

Polyphenols, minerals, fibers, and fruits hold significant potential for managing IBS symptoms and improving the overall quality of life of patients, as alternative or complementary therapies. Several studies have shown that polyphenols from curcumin, fennel, aloe, anise oil, peppermint oil, brewer’s yeast, and soy may be effective in improving IBS symptoms. Minerals, such as magnesium, zinc, and selenium, have also been shown to have potential benefits for IBS. Fibers can help regulate bowel movements and reduce constipation. Fibers, fruits, and polyphenols are closely related in terms of their contributions to a healthy diet and overall well-being. However, it is important to acknowledge the variability in individual responses to these dietary components and the need for personalized approaches. Further research is warranted to elucidate optimal dosages, combinations, and long-term effects, while also considering potential interactions with existing treatments or medications.

## Figures and Tables

**Figure 1 nutrients-15-04070-f001:**
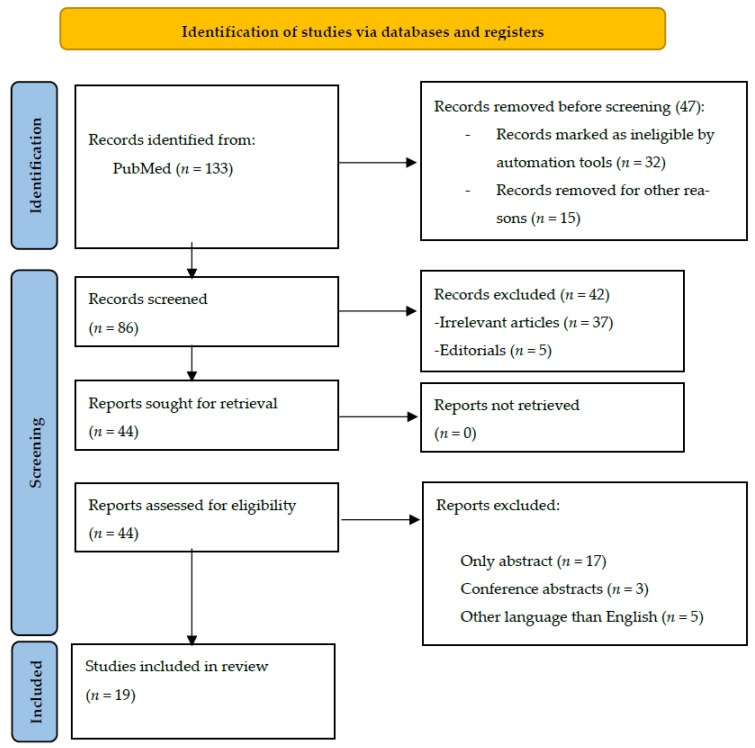
PRISMA flowchart of the included studies.

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
