# Peer review of "The Effect of Polyphenols, Minerals, Fibers, and Fruits on Irritable Bowel Syndrome: A Systematic Review"

_nutrients, 2023, doi:10.3390/nu15184070_

Round 1
Reviewer 1 Report
The review entitled “The Effect of Polyphenols, Minerals, Fiber, and Fruits in Irritable Bowel Syndrome: A Systematic Review” is interesting and has the potential to be published, but some points need improvement.
- The first point that stands out is that the authors did not cite the revision report in PROSPERO. There is also no recorded reference to a revision on this matter. I am not sure if MDPI requires registration, but the vast majority of journals that publish systematic reviews require deposit with PROSPERO for publication. Furthermore, which version of the PRISMA guidelines did the authors use in preparing the review?
- The authors mention that the following databases were searched: PubMed, MEDLINE, Embase, Scopus, Web of Science, and the Cochrane Library. However, only the pubmed search appears in the flowchart.
- The authors state in the flowchart that "records marked as ineligible by automation tools" which tools were used?
- Perhaps the most confusing part of the report is the description of the results.
The first item described in the results is 3.1 polyphenols, but the authors begin the description with xyloglucan. Is xyloglucan a polyphenol?
Shortly thereafter, curcumin fennel essential oil, aloe barbadensis mill, herbal combinations (curry, pomegranate and turmeric), peppermint oil and so on are described. In particular, I found this quite confusing since the studies cited examined essential oil, herbal combinations, etc., but not polyphenols. Did the studies described characterize the polyphenols contained in these products? If so, then the authors should provide a description of these results (chemical composition), since the item's title is "Polyphenols". If the studies described did not examine the polyphenols contained in the products tested, I suggest that the authors find a better way to divide the results found and the focus of the systematic review. For these studies to be included in the "polyphenols" item, the product tested would ideally have to be composed primarily of polyphenols or even be a polyphenol.
The authors also describe a study that investigated the effect of ginger in the treatment of irritable bowel syndrome. How was the study conducted? Did the patients consume the whole ginger? Was it a special preparation? This type of detail needs to be described for the reader for all studies.
The same is true for item 3.2 Fruits, fiber, and minerals.
- The discussion needs to be reconsidered and should include all studies. It seems that the authors focused a lot on polyphenols and peppermint oil and little on the other studies.
Author Response
We want to express our gratitude for your review of our manuscript. Your insights and recommendations were invaluable in improving the quality of our work. We carefully considered your suggestions and have made the necessary modifications to address the concerns you raised. We are truly appreciative of your time and effort in assisting us to refine our work.

Reviewer 2 Report
General comments:
The manuscript is a systematic review about the effect of different nutrients and foods in irritable bowel disease symptoms. The design of the work contains some inconsistencies in its orientation that make its results unclear. On the one hand, only English-language papers have been included in the review. It is a known fact that papers with negative results are more likely to be published in alternative languages. On the other hand, the work tries to cover too many different parameters: polyphenols, minerals, fiber and fruits. In some cases only one paper has been found that refers to them. On the other hand, some of the papers present doubts as to whether they have been correctly assigned to the group to which they have been assigned. Pipermint oil, or any other oil, it is more than doubtful that it can be considered as a source of polyphenols, since although it contains polyphenols, its main content is other and the effect obtained could well be due to other components. Fruits are reduced to kiwifruit, and minerals are considered to be represented by a diet with low starch and sucrose content, which also presents serious doubts as to their representativeness.
In short, I consider that the work carried out and the possible conclusions are too diffuse to be considered as cause-effect relevant to the pathology studied.
Specific comments:
Please avoid abbreviations from the abstract section. Particularly, no sense is found for the term “RCTs”, because it was not used after its abbreviation.
In the Introduction section, a very polyphenols have been given too much prominence, while the other components have been given too little attention. A better balance would be advisable.
Page 2, lines 92-93. “They can act as prebiotics….species”. A reference to reinforce with affirmation would be useful. In example Roca-Saavedra et al. Food additives, contaminants and other minor components: effects on human gut microbiota—a review.
Page 3, line 132: “Irritable Bowel Disease” was previously defined as “IBS”, is not necessary to cite it twice.
Page 3, line 142: “aged 18 years or older”. If the review was made only for adult people and not all type of patients, it should be specified in the title of the manuscript.
Page 3, line 148: “Meassures” should not be written with capital letters.
Page 5, lines 174-180. This information in not necessary or cited previously and can be deleted.
Tables 1 and 2. References included in the Tables also should be cited according to Nutrient´s instructions for authors. In the row “main outcome” a brief description of the findings is better than numeric values.
Author Response
Your expertise has undoubtedly enhanced the rigor and clarity of our research, and we are confident that these changes will significantly strengthen our contribution to the field. We are truly appreciative of your time and effort in assisting us to refine our work.

Reviewer 3 Report
Dear Editor,
thanks so much for the opportunity to revise the work entitled "The Effect of Polyphenols, Minerals, Fiber, and Fruits in Irritable Bowel Syndrome: A Systematic Review”.
The work is very interesting, evaluates the effects of polyphenols, minerals, fiber, and fruits on the symptoms and overall wellbeing of individuals with IBS. The work may serve as a reference for future research and development of diet recommendations and treatment of IBS.
The paper is well written, the results are clearly reported and the review of the literature rigorous.
I have not specific revisions for the authors to perform. I only suggest improving the introduction with more information about fiber and several minerals intake in IBS.
Thanks.
Author Response
Your insights and recommendations were invaluable in improving the quality of our work. We modified accordingly. Thank you once again for your invaluable guidance.

Reviewer 4 Report
Chiarioni and colleagues discuss the potential role of dietary factors, including polyphenols, minerals, fiber, and fruits, in managing Irritable Bowel Syndrome (IBS) symptoms. To systematically explore this, they performed a comprehensive review of several databases to gather relevant studies published up until July 2023. The accumulated evidence from these studies suggests that dietary interventions involving polyphenols, minerals, fiber, and fruits can have a beneficial influence on mitigating IBS symptoms. Specifically, the supplementation of dietary fiber, especially soluble fiber, was associated with a reduction in bloating and an enhancement in stool consistency. Despite the promising outcomes observed with these dietary interventions in alleviating IBS symptoms, further research is needed to establish their effectiveness and optimal usage. This review paper looks interesting to me. I am looking forward to looking at a revised version if the authors can address some major and minor issues satisfactorily. My comments are as follows:
Major issues:
1. I wonder what is the causal explanation for why the dietary intervention with dietary fiber can mitigate IBS symptoms. Summarizing causal reasons revealed or hypothesized by previous papers is very important to help readers understand the mechanism behind the beneficial effects.
2. How the supplementation of dietary fiber such as soluble fiber influences the gut microbiota and microbiota-derived metabolites is lacking. The authors need to carefully review those papers and investigate whether the changes in microbial composition or microbiota-derived metabolites induced by the dietary interventions contribute to alleviating IBS symptoms. Also, what is the implication of changes in the gut microbiota induced by dietary interventions involving polyphenols, minerals, fiber, and fruits? Does the gut microbiome change the metabolism and thus influence the host health (Tong Wang et al., PloS Computational Biology 2019)?
Minor comments:
1. Line 116: “a steppingstone towards” -> “a stepping stone towards”
2. Line 192: “irrelevant for our research theme” -> “irrelevant to our research theme”
3. Lines 239-240: “The study also recorded more adverse effects in the treatment group as in the placebo one” -> “The study also recorded more adverse effects than the treatment group as in the placebo one”
4. Line 268: “participants which have received peppermint oil” -> “participants who have received peppermint oil”
5. Line 458: “hold a significant potential” -> “hold significant potential”
6. Line 459: “improving overall quality of life” -> “improving the overall quality of life”
A handful of grammatical errors were detected.
Author Response
Thank you for your valuable feedback on our submission. We appreciate your recommendations and have made the necessary modifications accordingly. Your insights have improved the quality of our research, and we are grateful for your assistance.

Round 2
Reviewer 2 Report
The manuscript was improved with respect to its original version
Reviewer 4 Report
The authors have made changes to address my comments.